# Dysregulation of DNAM-1-Mediated NK Cell Anti-Cancer Responses in the Tumor Microenvironment

**DOI:** 10.3390/cancers15184616

**Published:** 2023-09-18

**Authors:** Rossella Paolini, Rosa Molfetta

**Affiliations:** Department of Molecular Medicine, Laboratory Affiliated to Istituto Pasteur Italia-Fondazione Cenci Bolognetti, Sapienza University of Rome, 00161 Rome, Italy; rossella.paolini@uniroma1.it

**Keywords:** NK cell immune surveillance, DNAM-1 activating receptor, DNAM-1 dysfunction, tumor microenvironment

## Abstract

**Simple Summary:**

Immune system counteracts tumor growth through a coordinated action of several innate and adaptative cells able to detect and eliminate altered cells. Among immune cells able to kill tumor cells, Natural killer (NK) cells belong to the innate arm of the immune system and distinguish cancerous from healthy cells thank to the expression of a wide array of activating and inhibitory receptors. DNAM-1 is an activating receptor that binds PVR and Nectin2 adhesion molecules frequently overexpressed on the surface of cancerous cells, thus representing a central receptor in tumor recognition. However, PVR and Nectin2 are also recognized by inhibitory receptors that are upregulated in tumor microenvironment and can counteract DNAM-1 activation, leading to NK cells hypo-functionality. This review focuses on the main potential molecular mechanisms responsible for the impairment of DNAM-1 functionality during tumor progression. Moreover, therapeutic approaches able to reverse DNAM-1 dysfunction and NK cell hypo-responsiveness will be also summarized.

**Abstract:**

NK cells play a pivotal role in anti-cancer immune responses, thanks to the expression of a wide array of inhibitory and activating receptors that regulate their cytotoxicity against transformed cells while preserving healthy cells from lysis. However, NK cells exhibit severe dysfunction in the tumor microenvironment, mainly due to the reduction of activating receptors and the induction or increased expression of inhibitory checkpoint receptors. An activating receptor that plays a central role in tumor recognition is the DNAM-1 receptor. It recognizes PVR and Nectin2 adhesion molecules, which are frequently overexpressed on the surface of cancerous cells. These ligands are also able to trigger inhibitory signals via immune checkpoint receptors that are upregulated in the tumor microenvironment and can counteract DNAM-1 activation. Among them, TIGIT has recently gained significant attention, since its targeting results in improved anti-tumor immune responses. This review aims to summarize how the recognition of PVR and Nectin2 by paired co-stimulatory/inhibitory receptors regulates NK cell-mediated clearance of transformed cells. Therapeutic approaches with the potential to reverse DNAM-1 dysfunction in the tumor microenvironment will be also discussed.

## 1. Introduction

Together with cytotoxic CD8^+^ T lymphocytes (CTL), Natural Killer (NK) cells exert a fundamental role against cancer development thanks to their ability to specifically induce lysis of transformed cells [1,2,3]. However, NK cells are innate lymphocytes that do not express an antigen-specific T cell receptor and have been recently classified as innate lymphoid cells (ILCs), which include five prototypical subsets that parallel the functions of CTL and CD4^+^ helper T cells [4]. Based on their functional properties and developmental trajectories, NK cells belong to group 1 ILCs together with ILC1: NK cells and ILC1 share the ability to produce Interferon (IFN)-γ, as well as the expression of the transcription factor T-bet [3,4]. Unlike the tissue-resident ILC1, NK cells are circulating cells that, in addition to cytokine production, also exert a cytotoxic function. Indeed, they release the content of preformed cytoplasmic granules against target cells, thus inducing perforin and granzyme B-mediated apoptosis of virus-infected and transformed cells [1,2].

In humans, peripheral blood NK cells are characterized by the surface expression of CD56 and the low-affinity FcγRIIIA receptor (CD16), and are classically split into two subsets: CD56brightCD16^−^ that produce high amounts of IFN-γ, and CD56dimCD16^+^ that are more cytotoxic [1,4].

NK cells activation is finely modulated by the integration of inhibitory and activating receptors [3,4]. Among activating receptors, Natural-Killer receptor group 2, member D (NKG2D), Natural cytotoxicity receptors (NCRs) and DNAX-associated molecule-1 (DNAM-1 or CD226) are mainly involved in tumor immune surveillance, since they bind self-proteins that are normally absent or poorly expressed on healthy cells and up-regulated upon tumor transformation. Their peculiarity is the ability to bind more than one ligand: NKG2D ligands include MHC-I related proteins, MICA/B and ULBP1-6 proteins in humans, and Rae-1α-ε, MULT1 and H60a-c in mice [5,6]; DNAM-1 ligands consist of two adhesion molecules, CD155 (PVR/Necl5) and CD112 (PVRL2/Nectin-2) [7,8]; NCR ligands include transmembrane, nuclear or soluble self-proteins as well as molecules derived from pathogens [9].

The main inhibitory signal for NK cells is provided by Major Histocompatibility Complex (MHC)-class I molecules. Receptors for these molecules include Ly49 receptors in mice or the Killer-cell immunoglobulin-like receptor (KIRs) in humans and CD94/NKG2A heterodimers. KIR and Ly49 receptors bind classical MHC-I molecules while CD94/NKG2A recognize human HLA-E and mouse Qa1. Upon ligand binding, these receptors trigger negative signals that prevent NK cell responses against healthy cells [10,11].

Down-regulation of MHC-I molecules (the missing self mechanism) and/or up-regulation of ligands for activating receptors (the induced self mechanism) frequently occurs in neoplastic cells, leading to the recognition and killing of target cells and the release of pro-inflammatory cytokines including IFN-γ and TNF-α [3,4,10]. In addition, CD16 binding to the Fc region of IgG results in perforin-mediated lysis of immunoglobulin-opsonized target cells through a mechanism named antibody-dependent cellular cytotoxicity (ADCC) [1,4]. Thus, NK cell infiltration in cancer is a generally favorable prognostic factor. However, it is likely that soluble factors present in the tumor microenvironment, including TGF-β, Indolemine-2,3-dioxygenase (IDO) and prostaglandin E2, or hypoxic conditions shape NK cell phenotype and functions. Indeed, tumor infiltrating NK cells can, in some cases, show pro-tumoral activity, characterized by decreased proinflammatory and cytolytic functions and the production of the pro-angiogenic factor VEGF [12].

Moreover, during tumor growth, NK cells progressively acquire a dysfunctional phenotype characterized by the down-regulation of activating receptors and the up-regulation of inhibitory receptors, which shift the balance toward suppressive signals and promote NK cell functional exhaustion [13]. Of note, on chronically activated NK cells, additional inhibitory receptors called immune checkpoint receptors can be up-regulated. They contribute to the acquisition of a dysfunctional phenotype and include not only programmed cell death protein-1 (PD-1), which is highly expressed on a subset of circulating NK cells [14], but also lymphocyte activation gene 3 (LAG3); T cell immunoglobulin- and mucin-domain-containing molecule 3 (TIM-3); T cell immunoreceptor with Ig and ITIM domains (TIGIT); and T cell-activated increased late expression (Tactile or CD96) [15].

Several features characterize NK cell dysfunction in the tumor microenvironment. NK cells decrease the expression of granzyme B and perforin as well as that of ligands for death receptors, including FasL and TRAIL. They also produce lower amounts of cytokines [16,17,18,19]. Adoptive transfer experiments in mice demonstrated that the NK cell dysfunctional phenotype is specifically induced through contact with transformed cells in the tumor microenvironment [20]. Conversely, the expression of activating receptors on the surface of NK cells is restored in patients who undergo remission [16].

A pivotal role in NK cell tumor immune surveillance is played by the DNAM-1/TIGIT/CD96/CD112R axis, a set of immunoglobulin immune receptors. This axis is characterized by the DNAM-1 activating receptor and the inhibitory receptors TIGIT, CD96 and CD112R, which share the ligands PVR and Nectin2 with DNAM-1, thus counteracting its positive action (Figure 1) [21,22].

In particular, DNAM-1 and TIGIT bind PVR and Nectin2 while CD96 recognizes PVR. Of note, both inhibitory receptors show a higher affinity than DNAM-1 for the common ligand PVR [23,24]. Moreover, the recent identification of PVRIG (CD112R), an inhibitory receptor for Nectin2, has introduced an additional level of complexity [25].

On NK cells, DNAM-1 recognition of its ligands initiates an activation signal that leads to the release of cytotoxic granules and the production of pro-inflammatory cytokines [26,27]. On CTL, DNAM-1 co-stimulates TCR activation, and its expression characterizes a more active subset compared with DNAM-1 negative/low T cells [28,29]. However, even though PVR and Nectin2 overexpression represents a danger signal that renders tumor cells susceptible to cytotoxic cell-mediated lysis, several mechanisms—including the increased expression of inhibitory receptors for these ligands—hamper DNAM-1 activation in advanced tumor stages [22].

This review recapitulates how NK cell function is regulated by the DNAM-1 receptor and its inhibitory counterparts, summarizing the therapeutic approaches that can be exploited to shift this receptor axis towards DNAM-1-mediated NK cell activation.

## 2. DNAM-1 Role in NK Cell Biology

DNAM-1 was originally described as a T-cell lineage-specific antigen (TLiSA-1), a specific marker of the differentiation of cytotoxic T cells [30]. Its role in the adhesion and cytotoxic function of NK and CTL was demonstrated years later by employing a specific monoclonal blocking antibody [31].

DNAM-1 is constitutively expressed on T and B lymphocytes and monocytes, and is a member of the immunoglobulin (Ig) superfamily, containing two Ig-like domains in its extracellular portion [21].

On cytotoxic lymphocytes, the interaction between DNAM-1 and its ligands promotes an activating signal that depends on its physical association with the integrin lymphocyte function-associated antigen 1 (LFA-1) [32]. DNAM-1 aggregation results in the protein-kinase C (PKC)-dependent phosphorylation of a serine residue on its cytoplasmic domain. This phosphorylation promotes the association with LFA-1 and the activation of the Fyn tyrosine-protein kinase that, in turn, phosphorylates a tyrosine residue in DNAM-1 cytoplasmic tail, thus initiating signal transduction [32,33,34,35]. Additional biochemical events triggered by DNAM-1 crosslinking have been described in mice, and a critical role for the cytoplasmic ITT domain has been envisaged [35]. Once phosphorylated, this motif associates with the adaptor Grb2 allowing the activation of Vav1, phosphatidylinositol 3′-kinase (PI3K) and phospholipase C-γ1. The same cytoplasmic domain is responsible for the activation of ERK1/2 and Akt serin/threonine kinases. All together, these molecular pathways promote actin reorganization that is required for both cytotoxic granule polarization and cytokine production.

However, on human NK cells, the simultaneous co-engagement of at least two activating receptors is required to induce cytotoxicity and IFN-γ production. In particular, DNAM-1 co-aggregation with 2B4 or NKp46 is necessary for the intracellular Ca^2+^ mobilization required for the initiation of the NK cell functional program [36]. The co-engagement of DNAM-1 with 2B4 enhances the tyrosine phosphorylation of adaptor molecule SLP-76 and the activation of exchange factor Vav-1, which overcomes signals by inhibitory receptors and triggers cytotoxicity [37].

DNAM-1 role in NK cell biology is not restricted to the activation of cytotoxic function but also involves maturation and education. NK cell maturation occurs in bone marrow, both in humans and in mice, from common lymphoid precursors that, through different intermediate stages, reach the optimal functional status before their egress from bone marrow and migration into the periphery [38,39]. During this process they undergo education, a process during which, through the recognition of self MHC-I molecules via their specific inhibitory receptors, NK cells acquire the ability to tolerate healthy cells [40]. The subset of NK cells that do not express inhibitory MHC-I-specific receptors are called uneducated and become hypo-responsive. Educated NK cells also acquire the ability to “sense” a decreased expression of the same MHC-I on infected or transformed cells, triggering cytotoxicity [10,40].

DNAM-1 is expressed early during murine NK cell maturation and its expression correlates with the NK cell’s ability to perform missing self recognition [41]. However, results obtained using a DNAM-1-deficient mouse model show that DNAM-1 presence is not required to induce education. Indeed, mature NK cells have the same ability to lyse MHC-I-deficient tumor cells as wild type mice [42]. DNAM-1 is present in about 50% of mature murine NK cells. Its presence alters NK cells ability to release cytokines and increases their proliferative potential. In particular, DNAM-1^−^ cells display reduced pro-inflammatory cytokine production, but higher chemokine secretion compared to DNAM-1^+^ cells [42].

On human NK cells, DNAM-1 expression is higher in educated compared to uneducated NK cells and correlates with the number of inhibitory receptors and the amount of cytolytic potential [43]. Upon target recognition, DNAM-1 facilitates the localization of the active form of LFA-1 integrin at the immunological synapse, promoting target cell killing [43].

Another important feature of NK cell biology is the ability of cytomegalovirus (CMV)-specific NK cells to expand in response to CMV infection. These NK cells are characterized by the expression of NKG2C receptor in humans and Ly49H in mice, and show some adaptative and memory-like phenotype in case of reinfection [44]. By using an anti-DNAM-1 blocking antibody and DNAM-1-deficient mice, a pivotal role for DNAM-1 expression in the expansion of CMV-positive cells was demonstrated [45]. Of note, the expansion of this subset upon transplantation into leukemic patients correlates with better prognoses and suggests a potential role for memory-like NK cells in anti-tumor responses [44]. However, whether DNAM-1 may contribute to the expansion or effector functions of this particular NK cell subset in tumors is still unknown.

## 3. DNAM-1 and Its Ligands: Regulation and Function in the Tumor Microenvironment

The two known ligands for DNAM-1, PVR and Nectin-2, are Ig-like transmembrane proteins involved in cell adhesion via homophilic and heterophilic interactions between nectin family members and/or components of the extracellular matrix (ECM) [7,8,46].

Although DNAM-1 interacts with both purified ligands with a comparable affinity in vitro, the strength of the interaction of DNAM-1 with membrane-bound PVR is higher compared to Nectin2 (Figure 1). Thus, PVR is considered the main ligand for DNAM-1 [8].

PVR and Nectin2 are poorly expressed on healthy cells and are restricted to a few cell types including spinal cord motor neurons, endothelial cells and some immune cells [47,48]. On antigen presenting cells (APC), the expression of DNAM-1 ligands can be induced by Toll-like receptor signaling through the activation of the NF-κB transcription factor [49,50]. This up-regulation promotes APC interaction with NK cells, thus regulating their maturation and polarization [49,51].

On tumor cells, several different molecular pathways are implicated in the transcriptional upregulation of DNAM-1 ligands, including DNA Damage Response (DDR) pathways, Sonic-Hedgehog signaling pathway, cytokine production and Fibroblast Growth Factor receptor stimulation [52,53,54,55,56,57].

Once transcribed, human PVR mRNA can be processed in different spliced variants, raising in four proteins that share the same extracellular domains: two soluble β and γ isoforms that are released in extracellular milieu, and two transmembrane α and δ isoforms that possess distinct cytoplasmic domains [58,59]. The expression of α and δ isoforms on the membrane of tumor cells may impact PVR function, since only α isoform can transduce intracellular signals thanks to the presence of an ITIM motif in its cytoplasmic domain. The ITIM confers to PVR the ability to trigger the activation of different signaling pathways leading to cell proliferation, inhibition of adhesion and the induction of cell migration, representing an intrinsic advantage for tumor growth and spread [60,61,62].

High expression of PVR and Nectin2 has been demonstrated on the surface of different solid and hematological human cancers [63,64,65,66,67,68,69,70,71,72], which become more sensitive to NK cell-mediated killing in vitro [63,64,65,66,67,68,69,70] and in vivo [67].

Several in vivo lines of evidence highlight the crucial role of DNAM-1 in tumor immune surveillance. Transplanted murine lymphoma cells were rejected more efficiently when transfected with PVR or Nectin2 [73]. In a genetic model of spontaneous B-cell leukemia, the activation of the DNA damage response during the early stages of tumor progression is responsible for the expression of PVR that, in turn, is sufficient to activate T and NK cell-mediated tumor regression [74]. Accordingly, DNAM-1-deficient mice showed increased tumor development and mortality after transplantation with tumors that express PVR, due to a reduction in the ability of NK and cytotoxic T cells to recognize and kill tumor cells [75]. These mice also reject chemically-induced cancers less efficiently than wild-type mice.

DNAM-1 role in tumor clearance appears to be particularly important for tumors that do not express other ligands for activating NK cell receptors (for example NKG2D), suggesting that the DNAM-1/DNAM-1 ligand axis extends NK cells capability to eliminate tumor cells [76].

In addition, DNAM-1-deficient mice are more susceptible than wild-type mice to the development of melanoma lung metastases as well as chemically-induced fibrosarcoma [24]. Moreover, in genetic models of spontaneous Multiple Myeloma (MM) development, a lack of DNAM-1 resulted in faster tumor progression and an impaired response to therapy with immune checkpoint blocking antibodies [77].

All together, these data demonstrate that DNAM-1 limits tumor development and progression in vivo.

However, several mechanisms may be responsible for the dysregulation of DNAM-1 activation in advanced tumor stages and will be discussed in the following subsections and paragraphs.

### 3.1. Ligand-Induced DNAM-1 Internalization Results in Impaired NK Cell Effector Functions

Upregulated NK cell receptor ligands on the surface of transformed cells may have paradoxical immune suppressive consequences. Indeed, several activating receptors are down-modulated upon a chronic engagement by their respective ligands expressed on tumor cells. The most striking example is given by NKG2D that is internalized and degraded in lysosomes upon sustained engagement both in human and murine NK cells [78]. Receptor down-modulation was also observed in NCRs in patients affected by myeloid leukemia, ovarian carcinoma and neuroblastoma [79,80,81,82,83].

Regarding DNAM-1, a reduction of its surface expression has been observed in NK cells from peritoneal effusions of ovarian carcinoma patients compared to their circulating counterparts [84]. This down-modulation is probably due to sustained interaction with PVR expressed on the surface of ovarian carcinoma cells, since co-incubation of peripheral blood NK cells with tumor cells reproduces DNAM-1 down-modulation [84]. In patients affected by different solid tumors, DNAM-1 down-modulation correlates with the activation of KIR receptor expression and impacts disease outcomes [85,86].

Moreover, compared to healthy donors, circulating NK cells derived from leukemic patients show reduced DNAM-1 expression that is likely induced by the interaction with its ligands [68,87,88].

Accordingly, in murine models of lung and breast cancer, increased expression levels of PVR on tumor cells is responsible for DNAM-1 down-modulation on both NK and T cells and correlates with tumor metastasization [57,89]. Mechanistically, in CTL, it has been formally demonstrated that upon PVR binding, DNAM-1 undergoes tyrosine phosphorylation in its cytoplasmic tail that promotes its ubiquitin-dependent internalization and proteasomal degradation [89]. It is likely that a similar mechanism is also operating on human NK cells, suggesting that DNAM-1 internalization may represent a mechanism responsible for NK cell dysfunction during tumor progression (Figure 2A).

### 3.2. Impairment of DNAM-1 Functionality by Altered Ligand Expression on Tumor Cells

Additional mechanisms responsible for DNAM-1 hypo-functionality in advanced tumors comprise post-translational ligand modifications or the release of ligands as soluble forms, with a consequent reduction of their expression on the surface of tumor cells (Figure 2B).

Post-translational protein modifications include reversible modifications, whereby ubiquitin or ubiquitin-like proteins are covalently attached to a substrate that subsequently becomes a target for proteasomal degradation or undergoes non-degradative functional alterations [90]. These pathways are often up-regulated in cancer and contribute towards modifying the cancer cell phenotype [91,92].

In hepatocellular carcinoma cells the activation of the unfolded protein response induces a constitutive PVR degradation [93], suggesting a role for the ubiquitin pathway. Our group showed the involvement of a different post-translational modification in MM cells, demonstrating that PVR is directly modified by the SUMO pathway [94]. This modification results in the intracellular retention of PVR and the reduction of its surface expression, leading not only to impaired NK cell immune surveillance, but also to reduced MM adhesion to bone marrow stromal cells [61,94].

Nectin2 surface expression also appears to be regulated by post-translational modifications. Indeed, a constitutive ubiquitination and proteasomal degradation render tumor cells, including MM, less susceptible to NK cell recognition and killing [95]. Accordingly, the therapeutic use of proteasomal inhibitors such as Bortezomib increases Nectin2 levels in this hematological malignancy [54,96].

All together, these data support a role for post-translational modification in reducing surface levels of DNAM-1 ligands, preventing NK cell recognition during tumor progression.

Unlike murine, human PVR is expressed not only as a transmembrane protein but also in a soluble form (sPVR) revealed in different body fluids such as blood, cerebrospinal fluid and urine [58,59]. Of note, DNAM-1 had a greater affinity than TIGIT and CD96 for sPVR [97], suggesting that the soluble ligand form of PVR preferentially bound to DNAM-1.

In patients affected by different epithelial cancers, the serum level of sPVR increases in relation to healthy donors and, in the case of gastric and breast tumors, correlates with disease progression [98,99]. Thus, high concentration of sPVR has been proposed as a marker of a poor prognosis.

Although it remains unclear whether increased production of sPVR represents a cause of cancer development, soluble ligand forms may act as a decoy protein, preventing the interaction of DNAM-1 with PVR-positive tumors. Accordingly, using lung metastasis models, Okumura and co-authors reported that sPVR inhibits DNAM-1-mediated NK cell cytotoxicity, exacerbating lung colonization by B16/BL6 melanoma cells [97]. It still remains undetermined whether sPVR affects not only NK cell function but also CTL anti-cancer activity. Moreover, how the expression of sPVR is regulated during tumor progression is still unclear.

## 4. DNAM-1 Dysfunction Caused by Inhibitory Checkpoints and Other Unrelated Receptors

Activated NK and CTL express a series of inhibitory receptors called checkpoints able to limit their function and prevent excessive activation. Of note, in the tumor microenvironment, these inhibitory receptors are over-expressed and render cytotoxic cells functionally defective in their immune surveillance activity [100]. In particular, the inhibitory checkpoint receptors TIGIT and CD96 are up-regulated during tumor progression and are able to compete with DNAM-1 for ligand binding.

Moreover, DNAM-1-triggered signal transduction may also be directly dampened by other unrelated receptors. These two additional mechanisms are both responsible for DNAM-1 hypo-functionality in the tumor microenvironment (Figure 2C) and will be further discussed.

### 4.1. DNAM-1 Inhibition by Checkpoint Inhibitory Receptors

The main inhibitory receptor that restricts DNAM-1 responses in the tumor microenvironment is TIGIT. TIGIT was initially identified on T cells as an inhibitory molecule able to promote immunoregulatory DC function [101]. Moreover, it can also exert an intrinsic inhibitory function on T cells, preventing their activation [102]. On naïve T cells, TIGIT is expressed only upon activation, while on NK cells it is constitutively expressed and its up-regulation in tumor-bearing mice and patients with colon cancer was associated with NK cell exhaustion [103].

TIGIT/PVR engagement can down-regulate both NK and T cell cytotoxicity [23,104,105]. TIGIT binding affinity for PVR is higher compared to DNAM-1, thus the main mechanism of DNAM-1 inhibition is the competition for PVR binding. TIGIT can also interact with Nectin2 but with lower affinity, thus its inhibitory action is probably predominantly exerted by PVR binding.

TIGIT contains an immunoglobulin domain in its extracellular region, and both an ITIM motif and one immunoglobulin tyrosine tail (ITT-like motif) in its cytoplasmic portion [106]. The ITIM domain, once phosphorylated upon TIGIT crosslinking, recruits the inositol phosphatase SHIP-1 that mainly prevents activation of the PI3K pathway [104]. In mice, besides the ITIM motif, the ITT domain is also required for inhibitory function [23]. Indeed, the ITT domain can recruit the cytosolic adaptor proteins growth factor receptor-bound protein 2 (Grb2) and β-arrestin2, both able to activate the inositol phosphatase SHIP-1 [107,108]. Through the recruitment of SHIP-1, Grb2 contributes to the inhibition of the PI3K pathway, but also to the dampening of the mitogen-activated protein kinase (MAPK) pathway, resulting in the inhibition of cytotoxic granule polarization and NK cell killing capability [107]. The recruitment and activation of SHIP-1 is also facilitated by β-arrestin2 and results in the inhibition of NF-κB activation and IFN-γ production [108].

Additional mechanisms of TIGIT inhibition have been described in T cells: TIGIT could directly interact with DNAM-1, reducing its ability to form homodimers and to signal. Moreover, it can induce inhibitory signals dephosphorylating DNAM-1 itself [29,109]. Even though it is not clear whether these latter mechanisms are involved in TIGIT-mediated inhibition of DNAM-1 in NK cells, a TIGIT blockade improves NK cell effector functions. A polyclonal anti-TIGIT antibody enhanced NK cell-mediated killing of tumor cells in vitro [23], demonstrating that TIGIT inhibition has the potential to improve NK cell anti-tumor functions.

During tumor progression, TIGIT is the main checkpoint responsible for functional exhaustion in NK cells [103]. Indeed, TIGIT deficiency or an antibody-mediated TIGIT blockade was sufficient to reverse NK cell exhaustion in murine models of subcutaneously implanted colon cancer, breast cancer, melanoma and chemically-induced fibrosarcoma, with the consequent impairment of tumor growth [103]. These results demonstrate that NK cells play a crucial role in responses to anti-TIGIT therapies.

Beside NK cells, several ex vivo studies in humans and the use of murine models have demonstrated that anti-TIGIT treatment can highly increase the efficacy of anti-PD-1 or anti-PD-L1 to reverse T cell exhaustion [109,110,111], explaining why the TIGIT/PVR axis raised great interest as a novel target for ICI treatments, mainly in combination with PD-1/PD-L1 pathway inhibition [112]. Several monoclonal antibodies blocking TIGIT interaction with PVR are in different phases of clinical trials and have already shown exciting results in non-small-cell lung cancer and melanoma [113]. However, their efficacy in hematological malignancies and other solid cancers is still unclear.

As an additional receptor of the DNAM-1 axis, CD96 interacts with PVR with an affinity that is intermediate between TIGIT and DNAM-1 (Figure 1) [114]. It is a transmembrane receptor belonging to the Ig superfamily and, besides NK cells, it is also expressed on activated T cells.

CD96 was initially found to facilitate adhesion between NK cells and their targets, thus favoring NK cell cytolysis [115]. However, CD96^−/−^ mice revealed an inhibitory role for CD96, since its absence improves IFN-γ production and reduces melanoma metastasis formation in the lungs [24]. Moreover, high CD96 expression correlates with poor prognoses in hepatocellular carcinoma patients [116]. Human CD96 can exert positive and negative actions on NK cell functions, since it contains in its cytoplasmic tail both the ITIM domain and the YXXM motif that can recruit the p85 subunit of PI3 kinase. This motif is absent in murine CD96 [117]. Moreover, human receptors can be expressed in different isoforms, raising from alternative splicing and displaying a different extracellular domain that results in variable binding affinity for PVR [118]. Therefore, even though CD96 function in NK cells is still debated, preclinical studies support the use of blocking anti-CD96 antibodies as a tool in anti-cancer therapy. Indeed, upon CD96 inhibition, NK cell anti-metastatic activity increased in murine models [119,120]. Of note, as observed in TIGIT, CD96 blocking may be combined with anti-PD-1 therapy administration. Indeed, treatment with anti-CD96 improves the efficacy of anti-PD-1 mAbs in a murine model of pancreatic cancer [121].

KIR2DL5 represents the most recently identified inhibitory receptor able to bind to PVR [122]. It is expressed on mature NK cells and belongs to the human KIR family. Interaction between KIR2DL5 and PVR occurs in a site of PVR that is distinct from the binding domain of DNAM-1, TIGIT and CD96. Thus, KIR2DL5 does not compete with other receptors for PVR binding. However, upon PVR binding, KIR2DL5 undergoes phosphorylation of ITIM and ITSM domains in its cytoplasmic tail, allowing the recruitment of Src homology regions 1 and 2 (SHP-1 and SHP-2) which dampen Vav1/ERK1/2/NF-κB signaling pathways. Of note, while DNAM-1 expression on circulating NK cells correlates with better prognoses in bladder cancer patients, KIR2DL5 expression worsens disease outcomes [123]. Accordingly, monoclonal antibodies inhibit KIR2DL5 interaction with PVR reduced tumor growth in several humanized tumor models [124], suggesting the blockade of the KIR2DL5/PVR interaction as a novel method of immunotherapy for treating human cancers.

DNAM-1 function can also be impaired by an inhibitory receptor that exclusively recognizes Nectin2 and that was originally named CD112R, and then termed PVRIG since it belongs to the family of PVR-like proteins [25]. It is composed of a single extracellular immunoglobulin variable-like (IgV) domain, a transmembrane region and a long intracellular domain containing an ITIM motif. Its expression is restricted to NK cells and effector/memory CD8^+^ T cells [25]. It is highly upregulated on the surface of tumor-infiltrating NK and T cells, especially in patients affected by solid tumors in areas including the breast, kidney, lung, prostate, ovary and endometrium [125,126].

CD112R binds Nectin2 with higher affinity compared to DNAM-1 (Figure 1), and dampens DNAM-1-mediated signaling by inhibiting the nuclear factor of activated T cells (NFAT) transcription factor [25]. Its inhibitory role in NK and CTL activation was further demonstrated by CD112R-deficient mice [127,128], and preclinical studies suggested the use of blocking antibodies, alone or in combination with TIGIT and PD-1 blockades, in anti-cancer therapy [125,127,128]. These promising results prompted the introduction of CD112R blocking antibodies in clinical trials.

It has recently been demonstrated in T cells that engagement of the checkpoint receptor PD-1 hampers DNAM-1 functions through the recruitment of SHP-2 phosphatase that dephosphorylates the DNAM-1 intracellular domain, thus interfering with DNAM-1-triggered signals (Figure 2D) [129,130]. Thus, PD-1-mediated inhibition represents an additional mechanism impairing DNAM-1 functions in murine tumor-infiltrating CD8^+^ T cells, and can be counteracted by treatments with anti-PD1 blocking antibodies [129]. Accordingly, the individual response to immune checkpoint inhibitor (ICI) therapy is largely dependent on the DNAM-1 expression levels in cytotoxic lymphocytes [28,89,130], highlighting a key role for DNAM-1 in the regulation of cytotoxic lymphocyte tumor immune surveillance. Whether this mechanism is also active in NK cells is currently unclear.

### 4.2. DNAM-1 Inhibition upon Chronic Stimulation of NKG2D

A recent report from our group has revealed a novel interplay between DNAM-1 and NKG2D. We have provided evidence that supports a role for NKG2D stimulation in dampening DNAM-1-mediated signaling. Indeed, we have demonstrated that upon sustained NKG2D stimulation with MICA in human NK cells, DNAM-1-triggered Pyk2 and ERK1/2 phosphorylation became defective, with a consequent impact on lytic granule polarization and the killing of PVR-expressing targets [131]. Even though the underlying molecular pathways are not completely clarified, this could represent an additional mechanism that dampens DNAM-1 function during tumor progression (Figure 2D).

Indeed, sustained NKG2D stimulation is a typical hallmark in the tumor microenvironment: high expression of NKG2D ligands promotes receptor down-modulation from the cell surface, with a consequent impairment of NK cell functions both in humans and in mice [78]. Accordingly, NK cell exhaustion does not occur in mice lacking NKG2D [132], supporting the notion that NKG2D is required for the induction of NK cell exhaustion in murine models. Moreover, in vitro NKG2D stimulation of the human NK cell is responsible for TIGIT upregulation [131,133], and this up-regulation represents an additive mechanism of DNAM-1-impaired activation. Further research is necessary to confirm these results in vivo and to clarify whether NK cell activating receptors other than NKG2D, such as NCRs, may play a similar role, being responsible for DNAM-1 hypo-functionality in the tumor microenvironment.

## 5. Concluding Remarks and Future Perspectives

Several reports have highlighted the importance of DNAM-1 activating receptor and that of its inhibitory counterparts in the regulation of NK cell-mediated immune responses. However, although accumulating results have clarified the molecular pathways that modulate the DNAM-1/TIGIT/CD96/CD112R axis in the tumor microenvironment, several questions remain unanswered. In particular, whether the expression of DNAM-1 ligands on tumor cells in vivo is beneficial or detrimental for tumor suppression is still unclear. An open question is whether PVR and Nectin2 interaction with DNAM-1 is useful only in the early stages of tumor development, before inevitably being completely overcome by inhibitory receptors in the more advanced and metastatic stages. This can open the possibility of therapeutic approaches intended to boost DNAM-1 ligand expression only during the early stages of tumor progression, while targeting checkpoint receptors may be indispensable at later stages. Another poorly defined aspect is the relative contribution of different inhibitory receptors to tumor progression. Moreover, whether the expressions of PVR and Nectin2 in humans have different clinical outcomes is largely unexplored. Of note, the presence of soluble PVR may exacerbate tumor progression through the selective inhibition of DNAM-1 functions. Understanding the mechanisms that regulate PVR alternative splicing in tumors may provide additional tools for therapeutic intervention.

In conclusion, DNAM-1 represents an important receptor in cancer immune surveillance, and new therapeutic approaches aiming to revert its dysfunction in the tumor microenvironment could result in the improvement of patient survival rates.

## Figures and Tables

**Figure 1 cancers-15-04616-f001:**
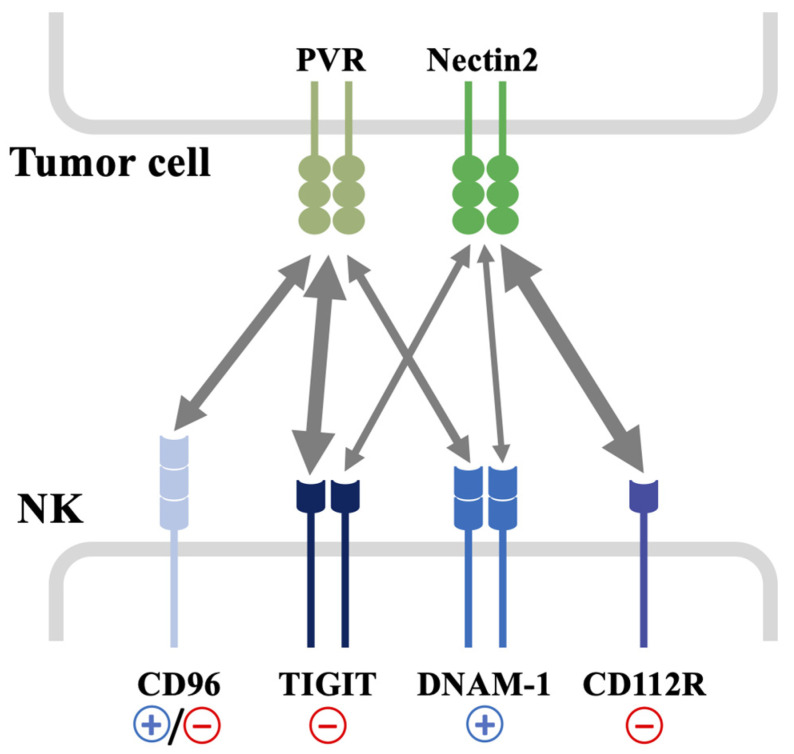
Schematic representation of different activating and inhibitory receptors sharing PVR and Nectin2 ligands. PVR and Nectin2 are frequently over-expressed on different types of tumors. They are both recognized with different binding affinities by DNAM-1, CD96, TIGIT and CD112R expressed on NK cells. The arrows indicate the functional interaction between the receptors and their common ligands promoting positive (+) or negative (−) signals, and the thickness of the arrows is proportional to the relative binding affinity. The strongest interactions are between TIGIT and PVR, and CD112R and Nectin2.

**Figure 2 cancers-15-04616-f002:**
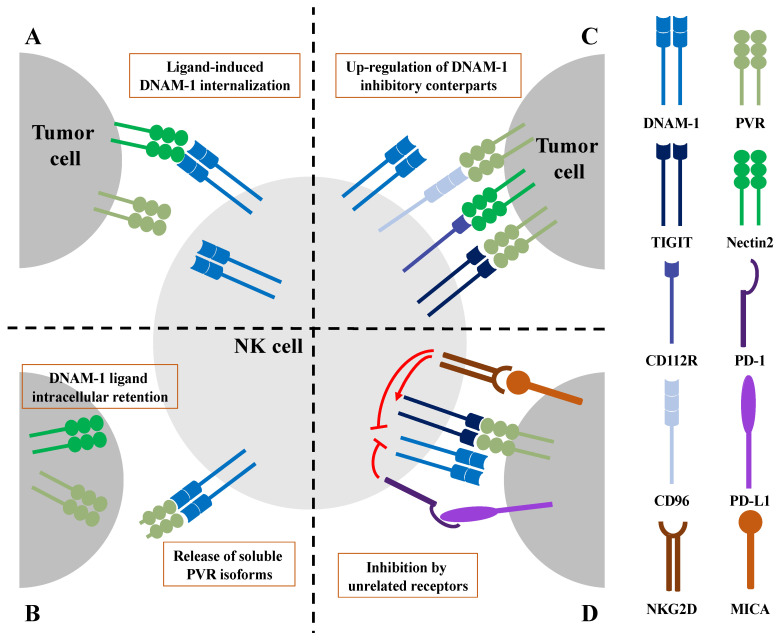
Different mechanisms of DNAM-1 dysfunction in the tumor microenvironment. (**A**) Chronic engagement of DNAM-1 with its ligands promotes receptor internalization and impairs DNAM-1-mediated functions. (**B**) Post-translational modifications are responsible for PVR and Nectin2 intracellular retention and/or degradation, thus preventing tumor cell recognition by NK cells. Soluble PVR isoforms can bind DNAM-1, interfering with recognition of membrane-bound ligands. (**C**) In advanced tumor stages, the up-regulation of checkpoint receptors competing with DNAM-1 for ligand binding hampers DNAM-1 anti-tumor functions. (**D**) PD-1 engagement is followed by dephosphorylation of DNAM-1 intracellular domains preventing signal transduction. NKG2D chronic engagement by tumor cells directly interferes with DNAM-1-triggered signaling and indirectly inhibits DNAM-1 activation by the up-regulation of TIGIT expression.

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
