# Peer review of "Dysregulation of DNAM-1-Mediated NK Cell Anti-Cancer Responses in the Tumor Microenvironment"

_cancers, 2023, doi:10.3390/cancers15184616_

Round 1

Reviewer 1 Report

I found the work really interesting and perfectly captures the important role that DNAM-1 plays in the education and response of NK cells and the mechanisms that tumors trigger to stop these mechanisms.

The work is very well written and reflects the current literature on the subject, although in my opinion it relies too much on results obtained in murine experimental models to demonstrate the importance of DNAM-1 in the anti-tumor response and how its down-regulation can reduce the antitumor activity of the host. There is literature that supports a similar pathway in humans in different types of cancer (MID: 31239317, PMID: 30242020), there are also works that explore the role of DNAM-1 ligands in the immune response mediated by cytolytic cells in pediatric leukemia (PMID: 34064810), or as KIR2DL5 appears to be directly involved in the susceptibility and severity of bladder cancer in humans (PMID: 31411976).

Perhaps the quality of the first figure could be improved a little bit?

Author Response

The work is very well written and reflects the current literature on the subject, although in my opinion it relies too much on results obtained in murine experimental models to demonstrate the importance of DNAM-1 in the anti-tumor response and how its down-regulation can reduce the antitumor activity of the host. There is literature that supports a similar pathway in humans in different types of cancer (MID: 31239317, PMID: 30242020), there are also works that explore the role of DNAM-1 ligands in the immune response mediated by cytolytic cells in pediatric leukemia (PMID: 34064810), or as KIR2DL5 appears to be directly involved in the susceptibility and severity of bladder cancer in humans (PMID: 31411976).

We thank the reviewer for her/his suggestion and added the requested references (72,85,86,123).

Perhaps the quality of the first figure could be improved a little bit?

We have improved the quality of the figure and changed the legend, accordingly.

Reviewer 2 Report

In the manuscript entitled “Dysregulation of DNAM-1-mediated NK cell anti-cancer responses in tumor microenvironment” the authors clearly highlight the key role of DNAM-1 in NK cells’ response against cancer. The review is interesting, clear and well organized.

COMMENTS:

Lines 56-58: When mentioning PD-1 expression on NK cells a fundamental reference paper is: Pesce et al 2017 (doi: 10.1016/j.jaci.2016.04.025)

Lines 125-128: When speaking about NK cell education also Ljunggren and Kärre work and hypothesis should be mentioned (Immunology Today 1990, 11: 237-244).

Line 160: The phrase: “… and are restricted in a few cell types …” should be changed to: “… and are restricted to a few cell types …”

Lines 278-279: The phrase: “Activated NK and CD8+ T cell express a series of inhibitory receptors called checkpoint receptor able to …” could be changed to: “Activated NK and CD8+ T cells express a series of inhibitory receptors called immune checkpoints able to …”

Line 327: ”Beside NK cells, several ex vivo studies in human …” should be: ”Beside NK cells, several ex vivo studies in humans …”

The manuscript is overall well written with only few grammar mistakes and typos.

Author Response

Lines 56-58: When mentioning PD-1 expression on NK cells a fundamental reference paper is: Pesce et al 2017 (doi: 10.1016/j.jaci.2016.04.025)

Lines 125-128: When speaking about NK cell education also Ljunggren and Kärre work and hypothesis should be mentioned (Immunology Today 1990, 11: 237-244).

We added the suggested references (14,10).

Line 160: The phrase: “… and are restricted in a few cell types …” should be changed to: “… and are restricted to a few cell types …”

Lines 278-279: The phrase: “Activated NK and CD8+ T cell express a series of inhibitory receptors called checkpoint receptor able to …” could be changed to: “Activated NK and CD8+ T cells express a series of inhibitory receptors called immune checkpoints able to …”

Line 327: ”Beside NK cells, several ex vivo studies in human …” should be: ”Beside NK cells, several ex vivo studies in humans …”

We thank the reviewer for her/his English editing and we have corrected, accordingly.

Reviewer 3 Report

Authors provided an analysis of the litterature to discuss how disregulation of the DNAM-1 molecule, accounting as a major NK cell activation rceptors, impacts on NK cell-mediated anti-tumor response.

The review highly focused on specific mechanisms governing NK cell activity, via DNAM-1 receptors, by exhaustively exploring the signaling pathway, the effectors and regulators involved and how tumor microenvironment cues orchestrate the normal and abnormal functions of NK cells in a DNAM-1 dependent manner.

I really appreciate the efforts, by the authors, to be very concise is dissecting a very complex topic.

I have only minor comments, as related to the introduction.

Since the audience of Cancers journal includes not only experts in tumor immunology, I think that a more detailed introduction on NK cell basic biology is necessary, to allow non-NK cells experts to better read/apperoack the review.  

Please consider including, very briefly, this bullet points in the introductions, to make the text more accessible to cancer biologists with poor immunology background:

1)    Short statement defining NK cells: innate immunity cell type, new classification  as type-1 ILC, origin, maturation, frequency in blood, major subset distribution.

2)    Short statement on their role in tumor as:

      -anti-tumor effector cells (ADCC, perforin, Granzymes, IFNg)

      -pro-tumor effectors cells (polarization state, pro-angiogenic activities,    immunosuppressive activities).

I would consider the review suitable to be accepted for publication, pending minor revision.

English uality is good.

Author Response

Since the audience of Cancers journal includes not only experts in tumor immunology, I think that a more detailed introduction on NK cell basic biology is necessary, to allow non-NK cells experts to better read/apperoack the review. 

Please consider including, very briefly, this bullet points in the introductions, to make the text more accessible to cancer biologists with poor immunology background:

1)    Short statement defining NK cells: innate immunity cell type, new classification  as type-1 ILC, origin, maturation, frequency in blood, major subset distribution.

2)    Short statement on their role in tumor as:

      -anti-tumor effector cells (ADCC, perforin, Granzymes, IFNg)

      -pro-tumor effectors cells (polarization state, pro-angiogenic activities,    immunosuppressive activities).

As suggested, we have corrected the introduction including additional details about NK cell biology and their functions during tumor progression.